# A Methodology of Policy Assessment at the Municipal Level: Costa Rica´s Readiness for the Implementation of Nature-Based-Solutions for Urban Stormwater Management

**Veronica Alejandra Neumann \* and Jochen Hack**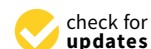

SEE-URBAN-WATER Junior Research Group, Section of Ecological Engineering, Institute of Applied Geosciences, Technische Universität Darmstadt, 64289 Darmstadt, Germany; hack@geo.tu-darmstadt.de

**\*** Correspondence: v.neumann@geo.tu-darmstadt.de; Tel.: +49-6151-16-20980

**Abstract:** Nature-based-solutions (NBS) pursue a combination of economic, social, and environmental benefits that can meet municipal goals on stormwater and rapid urbanization problems. However, NBS have fallen behind in reaching to the political and legal framework, and with this, to a policy mix for urban stormwater sustainability. When looking closer at NBS, it becomes evident that they are loaded with many barriers, including institutional and political ones, as well as those that exist in the urban area social context. These barriers are also deepened by the lack of policy guidelines and few demonstration projects. In this respect, this paper combines the concepts of urban experiments and the policy feedback cycle (PFC) into a singular assessment tool. It´s goal is to assess Costa Rica's municipal readiness in the implementation of NBS within the context of policy design and implementation. Therefore, this paper focusses on the first two stages of the PFC of an existing urban experiment to extract its policy insights for the successful replication of NBS projects. This novel method aims to contribute to the ongoing debate with respect to the ability of experimentation to prompt scalability and transferability of results. Hence, the New York City Green Infrastructure plan is considered as an urban experiment that promotes sustainable policy initiatives; while the PFC can identify and (re)formulate these policies initiatives and barriers into an adaptable policy guideline. Results indicate that sustainability policies at the municipal level should incorporate incentive mechanisms policies on (i) community involvement and communication; and (ii) transdisciplinary knowledge transfer between specialists and stakeholders. Finally, this paper suggests the inter-municipal cross-institutional collaboration and the recognition of external trigger events to incentivize a sustainable urban transition.

**Keywords:** nature-based-solutions; sustainable urban transition; policy feedback cycle; urban experiments; urban stormwater management

## 1. Introduction

It is becoming increasingly evident that current grey infrastructure systems are not always a sustainable solution for stormwater and rapid urbanization problems [1]. As grey infrastructures only fulfill single functions and put heavy burdens on governments, such as maintenance costs, it is more than obvious that a new solution mix is needed [2]. Part of this solution lies within the city itself. As urbanization continues to increase, governments need to ensure that citizens collectively continue to enjoy nature by living in harmony with their ecosystems, and with this, reducing the environmental and economic risks that climate change and loss of biodiversity implies [3]. For this reason, the recent

concept of nature-based-solutions (NBS) to address societal challenges, such as natural disasters and human well-being, has emerged.

The International Union for Conservation of Nature (IUCN) [4] defines NBS as: "actions to protect, sustainably manage and restore natural or modified ecosystems that address societal challenges by providing human well-being and biodiversity benefits". To that end, municipalities will need policy assessments of their existing sustainability policies to embed the NBS concept and to enable the necessary urban planning transition to take place [5–7].

This urban planning transition is understood as a reduction of dependence on traditional or grey infrastructure systems to the adoption of combined green and grey systems that provide additional ecological and social benefits. Grey infrastructure systems refer to engineered infrastructure, such as stormwater drainpipes, wastewater treatment plants, manholes, and to other components of a centralized water management system to remove on site stormwater and to discharge it to the nearest body of water, such as rivers [8]. While, the green infrastructure refers to the use of the natural processes such as infiltration and evapotranspiration, to reduce and slow down runoff, i.e., to mimic natural water balance processes, and thereby, to conserve or reestablish natural ecosystem values [9]. These ecosystem values are in the form of food, materials, clean air, pollination, etc. [10]. Such green infrastructure (GI) measures have been successfully applied in cities such as Malmö, Sweden [11], Copenhagen, Denmark [12], etc. This necessary urban transition to sustainable development is a major objective and driving force for policy design and urban planning [13]. Nonetheless, policymakers and stakeholders continuously focus on the design and function of traditional infrastructures and on the local top-down planning strategy, exacerbating with this, socio-economic disparities and unsanitary conditions [14,15]. In this context, sustainability policies for the implementation of NBS, such as green infrastructures (GIs), can contribute to urban planning transition in a way that is both participatory and inclusive [16–18]. Ideally, the transdisciplinary and multifunctional approach of NBS, which takes into consideration the participation of relevant stakeholders, can provide an opportunity for knowledge transfer from various sources (scientific and nonscientific) for the co-design of sustainability solutions [19]. Furthermore, with this they contribute to the political and social awareness of the benefits of ecosystem services for decision-making [20–22].

Additionally, sustainability policies for NBS can meet municipalities goals for housing, stormwater, recreation, etc., if they are placed in the context of other complementary local policies and programs [23–27]. In this regard, municipalities can start implementing urban experiment projects related to NBS to reduce difficulties in the reallocation of municipal funds due to the lack of policy guidelines [28]. These projects, as stated by the European Commission, can contribute to increase the political awareness and regulatory predictability of NBS [28,29]. Ideally, this demonstration effort will provide a basis for the embeddedness of the NBS concept into local municipalities' goals for urban stormwater management and rapid urbanization. Similarly, urban experimentation can signify an opportunity for different stakeholders to proof the performance and experience the benefits of NBS [28,29].

In this context, the policy assessment of existing urban experiments can raise political awareness for socio-economic and environmental arguments [9,30–32]. Here, experimentation in a real-world lab, as stated in the German context as Reallabore, as a leading approach in transformative and sustainable development research [33], has given rise to a new generation of experimental settings in the field of environmental politics [34]. Additionally, experimenting in a real-world setting can enhance sites of knowledge production for potential solutions for sustainability challenges [35]. Therefore, urban experiments understood as an experimental approach of transformative research can create a recursive and cyclical process of learning and data collection for policy making [36]. This process, analyzed through a policy feedback cycle (PFC), seeks to identify and assess policies in a cyclical manner to advice policymakers of future interventions [37,38]. In this regard, Chini et al. (2017) have used the PFC tool to evaluate GI experiments in 27 cities across the United Stated to obtain a large number of best management practices and policy insights [9]. That is, the PFC, which places current policies in a problem-oriented framework, can provide an opportunity to extract policy outcomes,

incentive mechanisms, and policy barriers from an urban experiment to promote its replication. Moreover, the PFC explores how policies "make citizens" and influence patterns of public and political support for the implementation of new innovations [39–43].

Consequently, the implementation of urban experiments can work as an evidence-based policy making. The new methodologies and policy tools used for the urban experiment contribute to the identification and (re)formulation of adaptable policies for its successful replication [9,18,44]. Similarly, it can provide the ground for inter-and trans-disciplinary knowledge transfer between different stakeholders enabling a more holistic approach for the co-design of municipal policies [29,45]. That is, urban experiments can complement to an urban planning transition by exposing innovations to real-world conditions for its further upscaling [9,46–48].

The present paper combines the concepts of urban experiment and PFC into a proposed policy assessment tool for the implementation of NBS. Its main goal is to assess the municipal readiness for NBS by assessing and applying the policy insights from an urban experiment to a specific case study with similar municipal and urban challenges. The premise of urban experimentation is based on the assumption that is possible to scale up from a demonstration project to other contexts through its processes of trialing and learning [49]. However, the knowledge produced in this manner tends to be highly contextual and difficult to transfer [50]. Hence, this paper aims to contribute to the ongoing debates in both the policy and academic world with respect to the ability of experimentation to prompt scalability and transferability of results, and with this to achieve a genuine urban transition [51–54]. Moreover, the proposed assessment tool offers a novel method of engagement and politics that can both challenge and complement current urban experimentation and policy processes and theories. It seeks to understand which social and material networks, in the context of existing socio-economic and political trajectories, affect a transition process. As a result, this paper focusses on the first and second stage of the PFC, policy identification and formulation, as a tool to identify and to assess the gap between the existing and the desired policy context for the implementation of future NBS projects.

In this perspective, the New York City (NYC) Green Infrastructure (GI) Plan [55,56] is considered as a guiding urban experiment that brings policy insights on the desired municipal context for NBS projects. While, the case study is in the Municipality of Flores in Costa Rica, which is one of the case studies of the transdisciplinary research group SEE-URBAN-WATER [57]. Essentially, this research group aims to study urban social-ecological problems in developing countries as a result of rapid urbanization and stormwater problems to stimulate natural alternatives and steer change [57]. Here, the application of the policy insights learned from the NYC GI Plan into the policy assessment of the Municipality of Flores can contribute to the inexorable question of how past experiments can generate scalable and transferable results to other contexts [51,58]. The Municipality of Flores is also located in a developing country with a leading policy framework for sustainability [59], yet with local political disputes on urban development and ecological topics [60]. Hence, the policy assessment of a municipality in Costa Rica can offer opportunities and instruments to induce controlled experiments in non-developed nations, and subsequently to capture political roles and social patterns to enable a cognitive break from the status quo to an urban sustainability transition.

The first part of this paper lays out the theoretical background of research and the methodology used to address the aims of the paper. The second part describes and analyses the results of the methodology applied to the case study. While, the third and fourth part correspond to the discussion and conclusion of the paper´s methodology and its results.

## 2. Background and Methodology

### 2.1. Nature-Based-Solutions for Urban Sustainability Policies

One aspect of urban sustainability comes in form of managing stormwater runoff, a challenging issue with the increasing amount of impervious areas in urban environments [9]. Forcing governments and engineers to plan and design cost-effective and holistic ways of addressing urban planning [61].

However, increased political and socio-environmental problems, combined with budgetary constraints are hampering the development of sustainable governance practices [22,62,63]. Similarly, most urban planners often fail to fully recognize the connection between humans and their natural surroundings [18, 64].

From the IUCN´s perspective, NBS and GI are terms that fit the most to tackle urban stormwater runoff problems [65–68]. The EU Commission [69] also defines green infrastructure as a strategically environmental network designed and managed to deliver a wide range of ecosystem services in both rural and urban settings. However, the wide-scale uptake of NBS, such as GI, have been hindered by the limited existing examples for evidence-based policy making and the overlapping concepts and terms used for the same purpose [19,29]. Thus, it is important that the NBS concept, which has emerged at the science-policy-practice interface, is placed in the context of existing terms and terminologies such as the one of GI. This can bring insights into a policy guideline of NBS that can guide its application and communication among policymakers, stakeholders and communities [19]. Similarly, this guideline can contribute to an urban transition from a complete grey stormwater system to an integrated green and grey system [9]. That is, to an urban stormwater sustainability transition.

### 2.2. Urban Experiments for Urban Sustainability Policies

Urban experiments are a relatively new area for the application and assessment of sustainability policy transitions [9,70,71]. Experimentation in an urban laboratory provides a mean to overcome policy challenges within the social, economic and environmental contexts. Thus, moving towards an urban planning transition with social interaction and sustainability goals [34,52].

Urban experimentation, as explored in the emerging literature, could be spontaneous or organized [44]. This experiment is designed to generate empirical evidence due to an intervention in order to foster sustainability transformations [72]. At this point, the literature on sustainability transition focuses on the city at a stage where urban experiments can contribute to policy making [73,74]. Much of this experimental stage is based on the premise that applying innovative solutions can effect broader transformative change in the city, and consequently, new policies initiatives for its replication and development [9,51].

### 2.2.1. The NYC Green Infrastructure Plan as a Guiding Urban Experiment for Urban Sustainability Policies

Voß et al. [5] stated that future efforts in sustainable transitions should include the results and procedures of experimentation. In this context, this paper considers the NYC GI Plan as a guiding urban experiment that can bring insights on potential policies and strategies needed for urban sustainability policies at the municipal level.

The aim of the NYC GI Plan is to provide a sustainable strategy to reduce urban stormwater runoff and contamination of waterways by using GI rather than the current all-grey strategy [55]. The success of this plan is mainly due to its continuous public engagement through various neighborhood association meetings, events kickoff and workshops about the design and financing of GI [75]. There has been a municipal and social interest to resolve stormwater problems where the Mayor of the city has been working closely with the United States Environmental Protection Agency (EPA), community leaders, environmental groups, and other key stakeholders to seek consensus on the scope of design and implementation of GI [76,77].

Additionally, the NYC Municipality has placed high importance in the recognition of the value of nature and the wider socio-economic and cultural benefits it implies [55]. This municipal interest combined with the pubic community outreach has enabled an inter-municipal cross institutional collaboration for the reallocation of funds for the implementation of GI [75]. Usually, the NYC DEP (Department of Environmental Protection) design, build and maintain the GI. While, the collaboration with other local departments, such as the Department of Education (DOE) among others, facilitate the research on grant programs, incentives and retrofit policies for GI [75]. As a result, the NYC GI plan has enabled new policies incentives for a stormwater management with a community and environmental

focus in NY. These policy insights and strategies used can work as an evidence-base policy making that encourage the replication of tools and methods for future NBS projects.

Particularly, some of the urban experiment projects of the NYC GI Plan have been implemented in low to middle-income neighborhoods with dense residential areas with limited green spaces [76]. For instance, one of the projects in the Queens neighborhood, a lower-income neighborhood [77,78], can bring general policy insights with regards to their transferability and scalability to other similar contexts; see Figure 1.

**NYC GI Plan in the Queens neighbourhood**

| GI goal | GI technology | Policy insights and strategy used |
|---------|---------------|-----------------------------------|
| - Infrastructure needed to absorb polluted stormwater and to prevent runoff<br>- Promote infiltration of stormwater runoff into the underlying soil | **Bioswales (right of way, ROW) on streets**<br>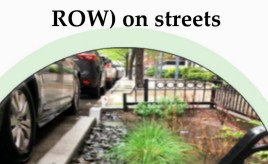 | - Enlist Business Improvement Districts and other community partners<br>- Create performance standard regulations for sidewalk reconstruction<br>- Integrate stormwater management into municipal capital programs<br>- Sewer charges for stormwater |
| - To promote infiltration<br>- To promote recreation on a public community asset that cultivates social resilience<br>- For aesthetic views value | **Rain gardens and permeable pavers on vacant lots**<br>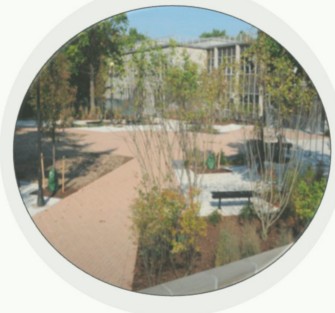 | - Rain garden stewardship program to partner with the community in maintaining the rain gardens.<br>- Local green infrastructure (GI) certification programs (training and examinations to local stakeholders)<br>- Zoning amendments for public space |
| - Promote clean air and reduce energy costs (heating or cooling)<br>- Rooftops stormwater retention and detention | **Green roofs at the hospital's main campus**<br>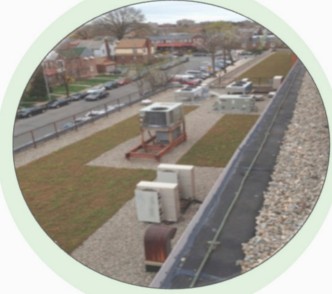 | - Regulations for stormwater performance<br>- Green roofs tax credit<br>- Green roof tax abatement<br>- GI grant programs for private property owners<br>- Housing preservation and development partnership for housing projects<br>- Community involvement |

**Figure 1.** Policy insights and strategies used by the NYC GI Plan at the neighborhood and site scale—at the Queens neighborhood located in the flushing catchment area of New York. The graph was constructed based on the information of the NYC GI Plan [76,79–82].

One of these contexts can be in the Llorente district, in the Municipality of Flores Costa Rica; see Figure 2, as it also ranks low in the human development index (HDI) [77,78,83]. The Llorente district and the Queens neighborhood are among the lowest-earning areas, with poor access to public transportation, schools and public and sanitary services [77,78,83]. Similarly, both of them have greater gender gap and low levels of neighborhood safety [77,78,83,84]. Suffering both of them from constant stormwater problems in the form of flooding, impervious surfaces and stormwater pollutants [85,86].

However, there are climatic differences between NYC and Flores. NYC´s climate is generally humid continental to warm humid subtropical in the Southeast and Long Island region. The last mentioned, has an annual average temperate of 12 °C and an average annual precipitation of 1200 mm [87]. In contrary, Flores has a tropical wet and dry climate with average annual temperatures of 22 °C and average annual precipitation of 1200 mm, most of it occurring during the wet season between May and October [88]. Both places have experience impacts from tropical cyclones and high intensity rainfalls. However, the differences in climatic conditions as well as site-and context-specific issues have to be taken into account when designing and implementing GI. Technical and ecological solutions from NYC cannot be replicated 1:1 In Flores. For this reason, SEE-URBAN-WATER is realizing empirical experiments to test the performance of adapted GI design in Flores [57].

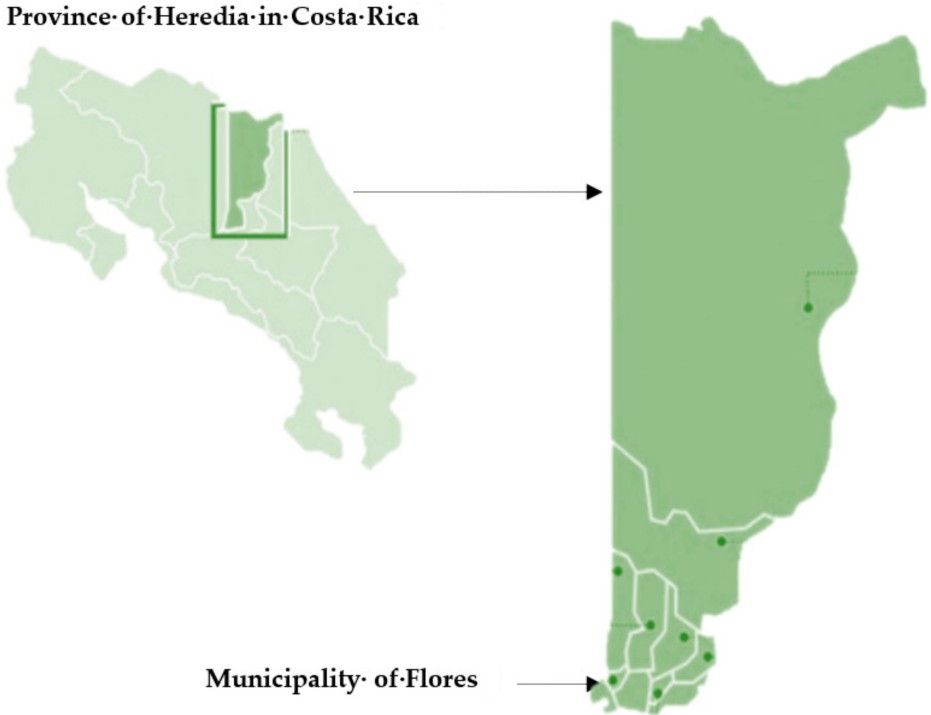

**Figure 2.** Map of Costa Rica with the indication of the Municipality of Flores [89].

From a policy standpoint, one of the main policy insights learned from this urban experiment, as seen in Figure 1, has been the need to reduce policy barriers and to increase socio-economic incentives for a successful implementation of GI; see Figure 3. Increasing incentives can allow municipalities to develop building and stormwater regulations to urbanization projects. While reducing barriers can allow investors and relevant stakeholders to re-evaluate the most efficient way to invest in ecological and infrastructure projects [56]. Some of these incentives mechanisms and barriers have been documented by the EPA on their Municipal Policy Handbooks to help local officials to implement GI in their communities [56].

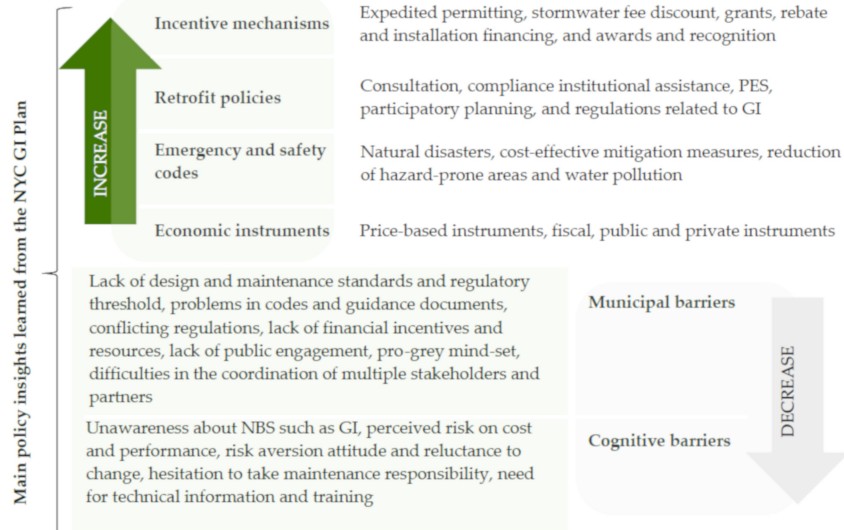

**Figure 3.** Policy insights learned from the urban experiment of the NYC GI Plan—increase incentive and reduce barriers for a sustainability urban policy mix. The graph was constructed based on the information of the NYC GI Plan and urban sustainability policies [76,79–82].

Moreover, these policy insights, to reduce policy barriers and increase incentives, represent a new approach to stormwater management that is not only sustainable and environmentally friendly, but inclusive as well. Many of these policies can be integrated within the context of several other complementary policies and programs. Hence, municipalities can apply GI policies to achieve both water-related goals and to tackle a host of other socio-economic and environmental problems [28,56]. This is possible through the identification of synergies in mission statements across local institutions, non-governmental organizations (NGO) and other agencies and social groups. For example, municipalities should consider the non-water benefits of NBS for energy conservation, public health, recreation, etc., to create constant communication and cooperation among stakeholders for overlapping priorities and goals. Consequently, the identification of incentives and barriers should go hand-in-hand with community involvement and inclusion processes [56]. Allowing a co-design approach for policy making.

### 2.2.2. A Future Urban Experiment Setting on NBS in the Llorente district, Municipality of Flores, Costa Rica

Costa Rica is recognized as a "green country" and global leader in promoting environmental policies [60]. It has a per capita income level ahead of the rest of countries in the region [90–93], and it highly aims to continue growing economically while managing natural resources efficiently and reducing pollution [94]. As a result, in 2015, Costa Rica pledged for an economy plan to decarbonize the country by 2050 [95]. According to the Minister of Environment, Carlos Manuel Rodríguez, one of the biggest aims are to cut the number of cars circulating in urban areas by 50% [96,97], and to expand its environmental policies, such as its well-known payment for environmental services (PES), to all municipalities [90,91,95]. However, until now, sustainable policies for urban planning are still a central topic of discussion for politicians [90,91]. Sustainability studies states that while Costa Rica remains an environmental leader in biodiversity and forest protection, it lags behind on wastewater management and public transportation [94]. The lack of wastewater treatment has meant Costa Rica´s demotion from one of the top 10 country´s performers in the environmental performance index (EPI) [94]. While, the gap in public transportation has increased the number of congested roads from private transport, harming the quality of life of many residential areas recently exposed to greenhouse gas (GHG) emissions [94]. Consequently, the lack of wastewater treatment and the gap in transportation have become major obstacles to green growth [94].

In this regard, municipalities find it difficult to reallocate municipal funds for alternative solutions in water and sanitation infrastructure without previous knowledge on its performance and benefits [94]. One of the biggest weaknesses in this sector has been the lack of pilot initiatives of alternatives solutions, and the late recognition of a political strategy on residential wastewater treatment [94]. Here, the water authority (AyA by its acronym in Spanish) recognizes the importance of designing and implementing new sustainable policies that protect Costa Rica´s natural resources, such as the water for consumption and sanitation; while mitigating the effects of climate change in the form of flooding [96,98]. Therefore, it it's possible to say that the application of urban experiments on NBS and the knowledge transfer of sustainability policies from past experiments can contribute to the reallocation of municipal funds for a sustainability transition.

In this context, a possible urban experiment setting on NBS in the Llorente district of the Municipality of Flores can be applied. This study area is located in the province of Heredia. A province which is highly affected for its pressing flooding and urbanization problems and for its political and social interest to solve them [99,100]. Heredia, located in the central-northern area, is also the smallest province of Costa Rica representing only 5.2% of the national territory and 10% of Costa Rica's total population [101,102].

Hence, the junior research group SEE URBAN WATER has identified three different scales to restore, alleviate or improve the urban water cycle in the Llorente District; while reducing the impact of rapid urbanization and climate change [57]; see Figure 4, and some potential experimental settings on GI which are still under analysis, see Figure 5.

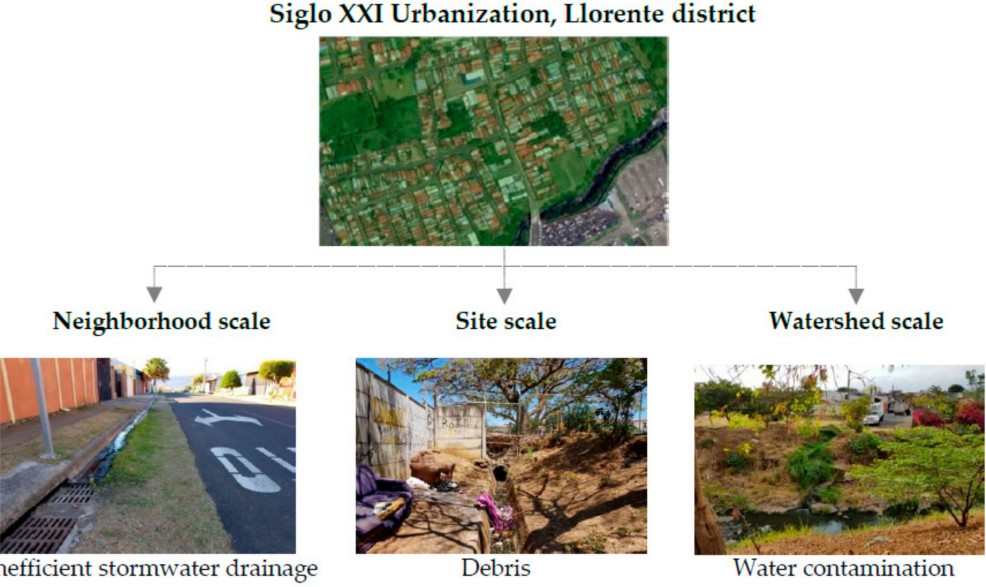

**Figure 4.** Scales for a future urban experiment settings on nature-based-solutions (NBS) in the Siglo XXI urbanization located in the Llorente district. These scales have been identified by the research group SEE-URBAN-WATER [57].

These three scales, as seen in Figure 4, are located in the Quebrada Seca-Rio Burío catchment in Costa Rica. The neighborhood scale is the area with considerable face-to-face interaction among other community members where all types of intervention affect its residents [57]; existing problems such as, current illegal discharges of grey water and direct discharge of stormwater runoff affect this scale. While, the site scale is an area located within the neighborhood with few interactions among the community member. The site scale tends to be on a free of building construction area where stormwater could be absorbed into the ground (infiltration) and to capture and reuse stormwater for residential purposes [57]. Finally, the watershed scale, which is the larger regional area, is the

interconnected natural network that provide essential ecosystem services [57]. Here, the watershed scale is located within the governance of several municipalities, requiring a greater effort on the part of the municipalities to carry out joint sustainability policies [57]. For this reason, this paper focus on the neighborhood and site scale as they are both located in the Municipality of Flores, where the case study is. Additionally, the policy assessment of these two scales can contribute to the design of a sustainable transition from a complete grey stormwater system to an integrated green and grey system within the municipal context.

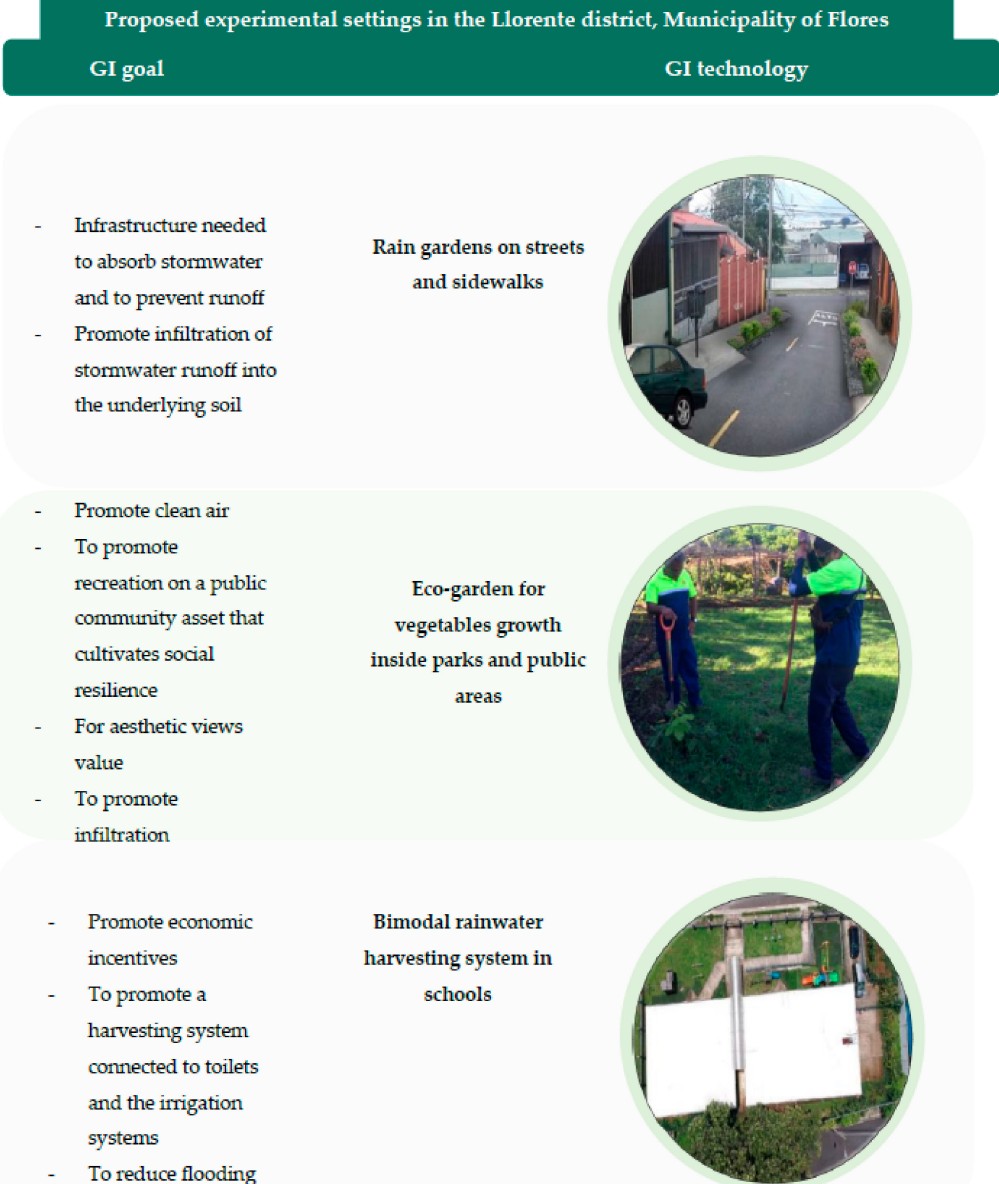

**Figure 5.** Proposed experimental settings on GI of the SEE URBAN WATER Junior Research Group [57].

In particular, it is important to mention that the future intervention, in the form of an urban experiment on NBS, in the Municipality of Flores is externally financed by the Federal Ministry of Education and Research of Germany [57]. This ministry aims to contribute to research programs for sustainable development in the area of socio-ecological research [103]. Consequently, the research group SEE-URBAN-WATER has been in contact with local counterparts to incentivize local engagement and policy insights for the implementation of alternative natural solutions [57].

*2.3. Methodology*

2.3.1. Proposed Methodological Tool to Endorse Sustainability Policies Related to NBS and Its Replication

This paper proposes a singular methodological tool to assess the policy readiness of the Municipality of Flores for the implementation of NBS. This tool is the result of a combination of two concepts: urban experiment and the policy feedback cycle (PFC). In this regard, this paper considers the NYC GI plan as a guiding urban experiment of NBS, from which its policy insights can be transferred, and the PFC of the European Geosciences Union (EGU) as the tool to extract such policy insights [104]. Additionally, the application of the PFC in this paper can contribute to enrich the policy assessment discourse as a helpful tool to identify and assess current policy frameworks [105]. As the PFC of the EGU also provides an opportunity to enunciate the distinct processes involved in policy identification, formulation, implementation and evaluation; see Table 1. Hence, the methodology of this paper aims to extract the policy insights for NBS from the NYC GI plan to assess its transferability and scalability to other contexts, such as the one in the Llorente district, Costa Rica; see Figure 6.

**Table 1.** Definition of all stages of the policy feedback cycle (PFC) as a policy assessment tool for the replication of NBS. The table was constructed based on the information of the PFC and urban sustainability policies [9,56,104–107].

| | Policy Stage | Definition |
|---|---|---|
| 1. | **Identification** | • Identification of urban development priorities and sustainability policies. In this case, existing and future sustainability policies should be able to gain greater community support if they address the needs and concerns of the community |
| 2. | **Design and Adoption** | • Analyze the gap between the existing and the desired sustainability policy mix for the implementation of urban priorities and policies <br> • Advise regulatory bodies from possible policy barriers and incentives |
| 3. | **Implementation** | • Establish policies and incentives or compliance assistance programs to increase funding, awareness and embeddedness of new concepts. This funding sources are in the form of policies on stormwater utility fees, flood control district funds, grants, etc. |
| 4. | **Evaluation** | • Assess the success of the policy, incentive or compliance assistance program |
| 5. | **Support and Maintenance** | • Modify, continue or expand policies based on the fourth stage <br> • Provide information to community and stakeholders |

Consequently, this paper focusses on the first and second stages of the PFC as they are key elements needed to proceed to the third stage of the PFC: the implementation of sustainability policies. Additionally, the first and second stages of the PFC have been already performed by the junior research group SEE-URBAN-WATER. Who has been constantly travelling to the study area since 2018 to identify urban development priorities and to engage stakeholders for the design and assessment of its current sustainability policies [57]. For this purpose, several workshops, bilateral meetings and field trips with relevant stakeholders and community regarding NBS have taken place [57]. It´s ultimate goal is to implement several NBS in the study area; while contributing to the municipal policy framework [57]. Therefore, the assessment of the first and second stages of the PFC of the NYC GI Plan and of the case study can contribute to highlight both the range of benefits that NBS provides and the holistic policy insights able to endorse its replication, and with this to its municipal policy readiness.

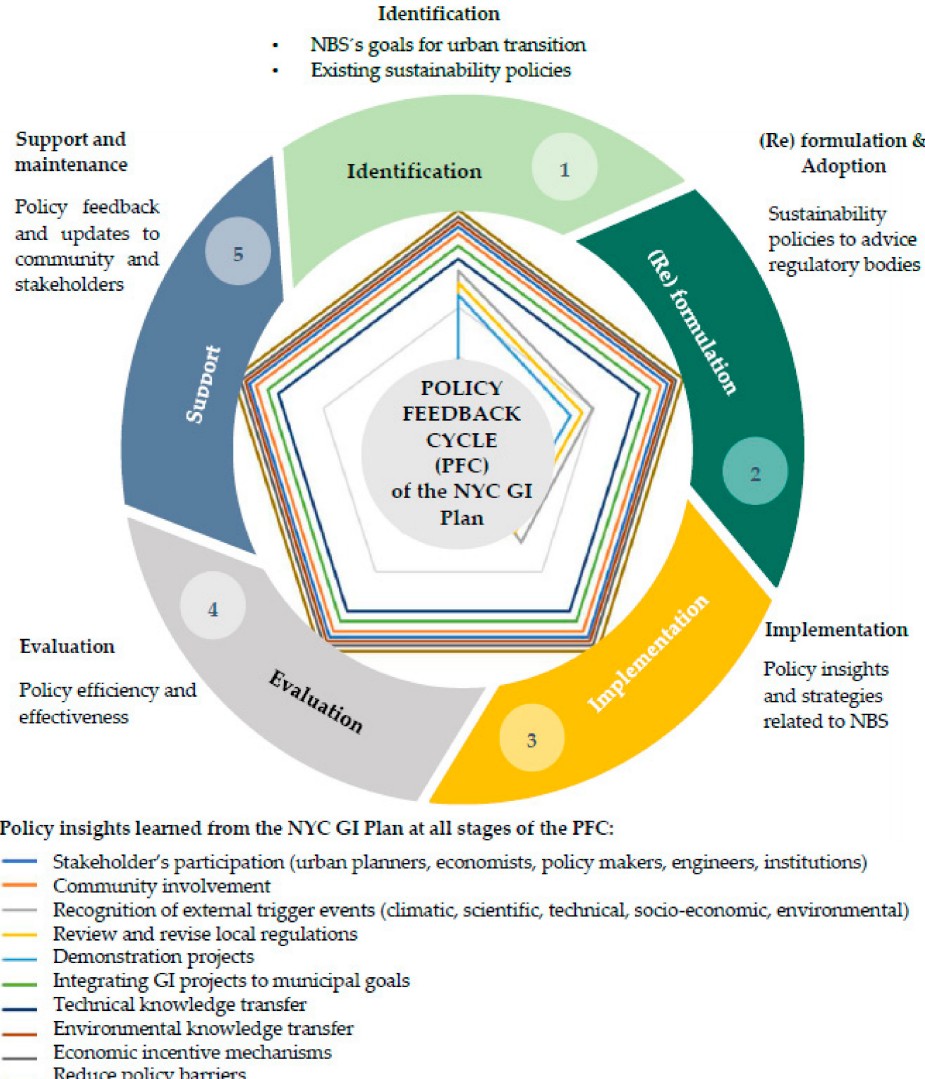

**Figure 6.** Policy insights learned from the PFC of the NYC GI Plan as a methodological tool to endorse sustainability policies related to NBS. The graph was constructed based on the information of the NYC GI Plan and urban sustainability policies [76,79–82,104].

2.3.2. Proposed Methodological Tool to Assess the Municipal Policy Readiness in the Implementation of NBS

Based on Figure 6, this paper elaborates a singular methodological tool, from the first and second stages of the PFC of the NYC GI Plan, to assess the municipal policy readiness in the implementation of NBS to other contexts; see Figure 7.

This tool does not only identify the most important policy insights of the NYC GI plan, but also the possible policy barriers and incentives for future NBS projects; see Table 2. Its objective is to translate social, economic, and ecological needs and concerns into policy terms, improving stakeholder awareness for urban sustainable solutions. Therefore, contributing to the environmental policy and urban development goal needed for an urban sustainability transition.

Furthermore, this proposed methodological tool spurs a shift in urban policy and planning discourse, in which the national top-down planning strategy is not any longer a central measurement for policy makers, but rather the constant stakeholder's participation and the recognition of external events for the co-design of subsequent policies outcomes. In this regard, the NBS concept can also be embedded into the existing policy mix by identifying and reducing the policy barriers. Thus, targeting the adoption of policies in the form of incentives or compliance assistance programs.

**Table 2.** Policy insights drawn from the first and second stages of the policy feedback cycle (PFC) of the NYC GI Plan at the neighborhood, site and watershed scale. The table was constructed based on the information of the NYC GI Plan and its urban sustainability policies [76,79–82,104].

| Policy Stage | Policy Insights | Policy Insights Assessment at the Neighborhood, Site and Watershed Scale |
|---|---|---|
| **1**<br>**Identification** | Urban development priorities | • Cleaning up NYC´s rivers, creeks, and coastal waters<br>• Improve water quality by achieving cleaner air and greener streets |
| | GI goals | • Reduce combined sewer overflows (CSO) volumes to optimize the existing wastewater system<br>• Improve water and air quality, help to cool the city, increase property values and beautify the communities<br>• Reduce energy bills and greenhouse gas emissions<br>• Control runoff from 10% of impervious surfaces through GI<br>• Institutionalize adaptive management, model impacts, measure CSOs, and monitor water quality<br>• Engage and enlist stakeholders |
| | GI technologies | • Advanced street-tree pits, permeable pavement, green and blue roofs, bioswales right of way (ROW), raingardens, constructed wetlands, rain barrels or cisterns, rooftop detention, etc. |
| | Sustainability policies | • Identify policies related to improve water and air quality, reduce energy bills and greenhouse gas emissions |

**Table 2.** *Cont.*

| Policy Stage | Policy Insights | Policy Insights Assessment at the Neighborhood, Site and Watershed Scale |
|---|---|---|
| **First and second stages Additional Policy insights and strategies used by the NYC GI Plan** | **Recognition of external trigger events** | • *Climatic*: there is the need to further reduce CSOs that discharge a mixture of untreated sewage and stormwater runoff when it rains<br>• *Scientific/technical advances*: New York City's Green Strategy is nimble enough to incorporate new natural technologies and approaches<br>• *Socio-economic*: The city is facing economic challenges and constrained resources, while the cost of grey infrastructure is significantly increasing and its marginal contribution to overall water quality diminishing. At the same time, residents want sustainability benefits such as increasing air quality, health and recreational activities |
| | **Stakeholder's participation and community involvement** | • Constant communication and meetings organized by the New York Department of Environmental Protection (DEP) with the Major of the City of New York, watershed partners, community, environmental groups, city agencies, urban planners and policymakers |
| | **Environmental and technical knowledge transfer** | • Methodological guidance and knowledge transfer to urban planners and community for the infrastructure installation and maintenance. The benefits of such infrastructures were also emphasized. DEP also provide resources and technical support so that communities can propose, build and maintain GI.<br>• Environmental and economic knowledge transfer through environmental impact assessments. The NYC GI plan elaborated impact assessment of its existing grey system and its hydraulic capacity vs. cost-effective green infrastructures systems |

**Table 2.** *Cont.*

| | |
|---|---|
| **Review and revise local regulations to overcome policy barriers** | <ul><li>SPDES (state pollutant discharge elimination system) general permit for stormwater discharges from construction activity [108]</li><li>SPDES multi-sector general permit for stormwater discharges associated with industrial activity [109]</li><li>Municipal separate storm sewer systems [110]</li><li>Protection of certain streams; disturbances of stream beds; permit, Environmental Conservation Law § 15-0501 [111]</li><li>Protection of water bodies; permit, Environmental Conservation Law § 15-0503 [112]</li><li>Powers and duties of the department, Environmental Conservation Law § 15-1303 [113]</li><li>National Environmental Policy Act (42 U.S.C. §§ 4321-4370a) [114]</li><li>State Environmental Quality Review, Environmental Conservation Law, Article 617 [115]</li><li>Waterfront Revitalization and Coastal Resources Act. Article 42 [116]</li><li>New York City Zoning Resolution [117]</li><li>NYC Building Code [118]</li></ul> |
| **Integration of GI projects into municipal goals** | <ul><li>Sewer and wastewater treatment plant upgrades related to the $1 billion in contracts for construction projects [119]</li><li>OneNYC 2050: to build a strong and fair city confronting climate crisis, equity and strengthen the democracy [120]</li></ul> |
| **Implementation of small-scale demonstration projects to highlight policy barriers and socio-economic incentives** | <ul><li>*Develop demonstration sites and large-scale experiments to demonstrate the relevance of GI.* DEP is building more than 20 demonstration projects in collaboration with other agencies and local authorities, including New York City Department of Parks and Recreation (DPR), etc. Some of these demonstrations include: green roofs, tree pits, and constructed wetlands, among others.</li><li>*Policy insights for an urban sustainability transition.* Policy insights learned from the urban experiment to increase socio-economic incentives and to reduce policy barriers; see Figure 3.</li></ul> |

**Table 2.** *Cont.*

| 2<br>**(Re)formulation and adoption** | Advice to regulatory bodies<br>*Bridging the gap between current and desired policy context that promote GI.* Reduce policy barriers and increase economic incentives mechanisms through the application of sustainability policies from the EPA (United States Environmental Protection Agency) | Suggested municipal policies and incentives for an urban sustainability transition:<br><br>• **Subsidy:** more funding for subsidy programs related to GI. This funding can come from stormwater fees collected by the Municipality<br>• **Consultation:** Provide free consultations with GI professionals to allow property owners to overcome initial uncertainty<br>• **Stormwater fee reduction:** To qualify to this credit the property owner must demonstrate that stormwater from his property is controlled with an on-site GI<br>• **Compliance assistance:** Inter-municipal cross department collaboration between the full spectrum of stakeholders<br>• **Reimbursement or free disconnection:** Property owners wanting to disconnect their own downspouts<br>• **Permeable pavement retrofit policy:** Private and public paved surfaces policies to manage stormwater and recharge groundwater. Programs such as the Green Alley Program of Chicago [121]<br>• **Regulations on impervious coverage within the watershed:** There shall be no new impervious ground cover constructed within 60m of the bank of a surface water body<br>• **Bioretention retrofit policy:** To treat runoff from private property in the form of incentives, such as grant programs. Additionally, to treat runoff from the public right-of-way in the form of regulations, such as easements to maintain rain gardens<br>• **Voluntary offset programs:** To implement rain barrels and rain gardens on private property at a low cost<br>• **Green lot retrofit policy:** to reduce toxic chemicals, criminal activity and increase property values, infiltration and recharge groundwater<br>• **Job creation:** develop new business and investment models and legal and institutional frameworks for GI. This contributes to a reconnection of citizens with nature to enhance their well-being<br>• **Co-design:** Green Infrastructure Task Force to design, build, inspect and control the implementation of the infrastructure and communicate it to policymakers |
| --- | --- | --- |

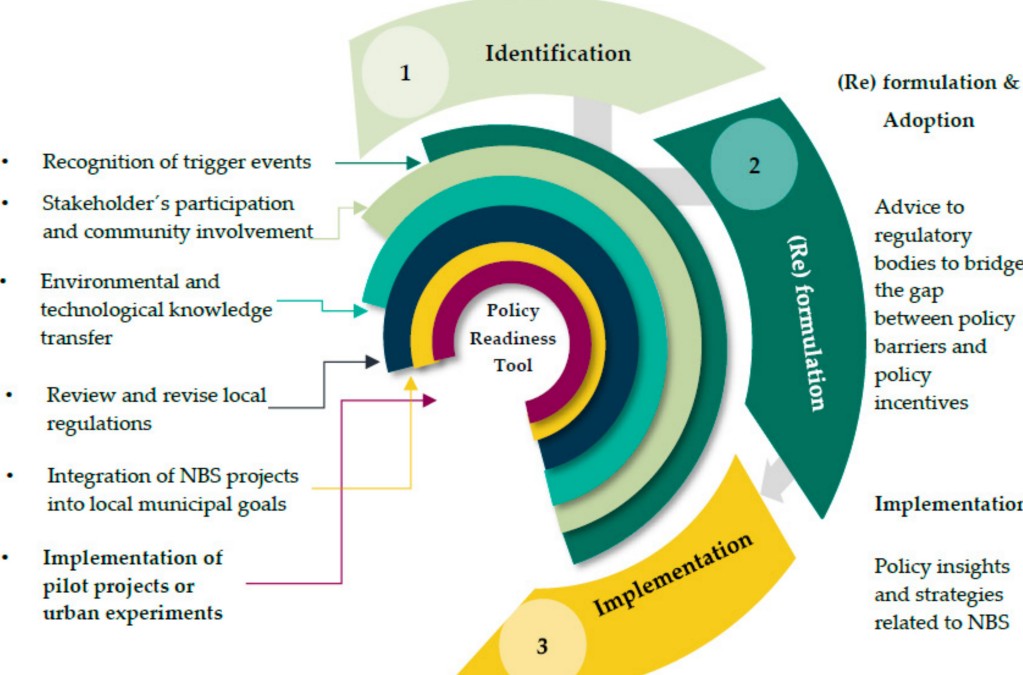

**Figure 7.** Proposed methodological tool to assess the municipal policy readiness in the implementation of sustainability policies related to NBS.

### 2.3.3. Data Collection

The preliminary desk-based research utilized database sources such as Scopus and Web of Science. The keyword search consisted on keywords from the title and abstract. From this initial selection of literature, articles according to the exclusion and inclusive criteria were excluded or included. Most of the excluded articles were published before 1996, while articles that were more relevant to policy assessment and urban sustainability were screened as relevant ones.

Alongside the urban experiments and PFC literature review, this paper also explores urban sustainability policies in national and municipal frameworks, in Costa Rica and NY, in which regulations, laws and policies relevant to the water management sector were studied.

In the case of Costa Rica, more than 45 international environmental treaties between 1990–2015 have been signed [122]. In addition, numerous regulatory policies in the environmental and biodiversity context have been enacted. Mostly, these policies are related to forest protection and to its ecosystem services than to water management and urban development [122]. As a result, this paper only analyzes the national and municipal regulatory policies related to water, health, and the environment within the urban planning context. Policies within the municipal context, such as regulatory plans for sustainability and municipal codes were studied more in detail. Here, it is worth mentioning that the Municipality of Flores, in the province of Heredia, is one of the only three out of the ten municipalities of the province with a regulatory plan [123]. The regulatory plan is a law that orders the urban development of the municipality to guarantee a safe and pleasant place to live; while identifying and finding local solutions, such as reducing flooding and facilitating the rehabilitation of urban areas [123].

Similarly, in New York only the regulatory policies, related to water, from the United States Environmental Protection Agency (EPA) [124] and from the NY Department of Environmental Protection were analyzed [125].



## 3. Results

*Policy Readiness Assessment of the Municipality of Flores in the Implementation of NBS*

In order to assess the municipal policy readiness for the implementation of NBS (third stage of the proposed methodological guideline; see Figure 7), it is in the first place important to identify and assess the urban development priorities and existing policies (first stage), and then proceed to bridge the gap between the current and the desired policy context (second stage); see for the results of these two steps in Table 3., while, applying the policy insights learned from the NYC GI plan; see Figure 7.

Transferring the policy insights of the NYC GI Plan to the current policy feedback cycle (PFC) tool yields seven additional policy insights for the successful implementation of NBS; see Figure 7. These policies insights are: (i) stakeholder´s participation; (ii) community involvement; (iii) recognition of external trigger events; (iv) review and revise local regulations; (v) integrate NBS projects into inter-municipal cross-departmental goals; (vi) technological knowledge transfer and (vii) implementation of demonstration projects. These policy insights are integrated in the first and second stages, policy identification and adoption, as key stages needed to assess the readiness for the implementation of successive policy outcomes.

Overall, applying the proposed policy readiness tool to the case study in the Municipality of Flores highlights a series of policy suggestions to take into consideration, see Table 3. Results suggest that the Municipality of Flores should include the following policies at the different scales for the implementation of NBS: (i) incentive mechanisms policies, such as grant programs, rebates and tax abatement; (ii) retrofit policies, such as compliance institutional assistance policies and urban PES; (iii) emergency and safety codes; (iv) participatory planning and governance; (v) regulations on biodiversity protection and amount of green spaces within the urbanization; and (vi) fiscal instruments such as policies that incentivize nature-affine investment from public authorities. These suggested policies aim to contribute to a more comprehensive system of environmental management by establishing environmental criteria for urban sustainability. Therefore, sustainability policies related to NBS at the municipal level would need to account for the fluid understanding of these measures and a continuous social communication to promote the widespread of NBS.

**Table 3.** Policy readiness assessment of the Municipality of Flores in the implementation of NBS.

| Stage | Policy Insights | Policy Insights Assessment at the Different Scales; See Figure 4. | |
|---|---|---|---|
| **First stage Identification** | Urban development priorities | ✓ | **Neighborhood scale:**<br><br>• Reduce illegal discharges of greywater, accumulation of pollution, and direct discharge of stormwater runoff<br>• Better distribution of green spaces and sidewalks design |
| | | ✓ | **Site scale:**<br><br>• Regeneration of previously neglected areas considered as dangerous and non-recreational<br>• Reduce accumulation of pollutant loads, illegal deposits of garbage and debris, and high degree of erosion and sealing |
| | | ✓ | **Watershed scale:**<br><br>• Reduce flooding, loss of habitat, and deterioration of the ecosystem |
| **First stage Identification** | NBS goals | ✓ | **Neighborhood scale:**<br><br>• Increase infiltration and recharge groundwater<br>• Reduction of pollutant levels<br>• Reduction of the urban heat island effect<br>• Rainwater harvesting for different uses, such as car wash<br>• Enhancing sustainable urbanization so that they are more attractive and safe<br>• Raising awareness of local products by using urban gardening |
| | | ✓ | **Site scale:**<br><br>• Increase air quality and property values<br>• Strengthen community cohesion by reconnecting people with nature<br>• Jobs creation by leveraging new business opportunities<br>• Reduction of costs associated to heat stress |
| | | ✓ | **Watershed scale:**<br><br>• Restore degraded ecosystems enabling them to deliver ecosystem services, and thus, socio-economic benefits<br>• Improve risk management and resilience by restoring its hydrology water storage |

**Table 3.** *Cont.*

| Stage | Policy Insights | Policy Insights Assessment at the Different Scales; See Figure 4. | |
|---|---|---|---|
| | Possible NBS technologies | ✓ | **Neighborhood scale:**<br><br>• Green roofs<br>• Bioswales right of way (ROW)<br>• Raingardens<br>• Rain barrels |
| | | ✓ | **Site scale:**<br><br>• Constructed wetland<br>• Parking lots with permeable paving |
| | | ✓ | **Watershed scale:**<br><br>• Habitat corridors |
| | Sustainability policies | ✓ | **Neighborhood, site and watershed scale:**<br><br>• Laws and regulations related to stormwater management and protection of natural resources; see Table A1 in the Appendix A [126–132]<br>• Municipal code No.7794 [133]<br>• Urban planning law of Costa Rica No.4240 [134]<br>• Regional urban development plan of the Greater Metropolitan Area (GAM) [135,136]<br>• Technical standards for design and construction of potable water, sanitation and rainwater supply systems [137]<br>• Regulatory plans of the Municipality of Flores [138,139] to promote a sustainable development in accordance with the real needs, capacities, and aspirations of its inhabitants. This plan regulates the following subsections:<br><br>  - Zoning ordinances: define compatible human activities in various zones [140]<br>  - Roads and transportation: provide regulations to solve the main road problems with the pedestrians as main priority [141]<br>  - Construction: provide regulations that guarantee the development of buildings in harmony with the environment [142]<br>  - Fractionation, urbanizations and condominiums: regulations to determine urban development requirements for the urbanizations [143]<br>  - Urban renewal: define policies to develop and recover areas in the canton [144]<br>  - Law for integral waste management No.8839 [145]<br>  - Law No. 8932 exemption from the payment of taxes on wastewater treatment system to help mitigate the contamination of water resources and improve water quality [146] |

**Table 3.** *Cont.*

| Stage | Policy Insights | Policy Insights Assessment at the Different Scales; See Figure 4. | |
|---|---|---|---|
| **First and second stages Additional policy insights and strategies used by the NYC GI Plan** | **Recognition of external trigger events** | ✓ | **Neighborhood, site and watershed scale:**<br><br>• Climatic: reduce flooding<br>• Scientific/technical advances: need to incorporate natural approaches to stormwater management<br>• Socio-economic: reduction of costs associated with floods and stormwater. Positive impacts on perceived health and wellbeing |
| | **Stakeholder´s participation and community involvement** | ✓ | **Neighborhood, site and watershed scale:**<br><br>• According to the regulatory plan of the municipality, Article 18 and 19 [139] the Strategic Board of Territorial Planning, JEPT for its acronym in Spanish, counts with a participatory urban planning strategy: a bottom-up model that emphasizes in involving the entire community for decision-making |
| | | × | **Neighborhood, site and watershed scale:**<br><br>• There is the need to expand the Article 18 and 19 of the regulatory plan of the municipality to include the wide spectrum of stakeholders for decision-making. |
| | **Environmental and technical knowledge transfer** | ✓ | **Neighborhood, site and watershed scale:**<br><br>• SEE-URBAN-WATER research group [57] provide methodological guidance and knowledge transfer to the municipality and community through workshops and bilateral meetings where the benefits of such infrastructures and social interests are emphasized |
| | | × | **Neighborhood, site and watershed scale:**<br><br>• There is the need to elaborate impact assessments of the existing grey system and its hydraulic capacity vs. cost-effective NBS. This can be in the form of cost-benefit analysis (CBA) |

**Table 3.** *Cont.*

| Stage | Policy Insights | Policy Insights Assessment at the Different Scales; See Figure 4. | |
|---|---|---|---|
| | **Review and revise local regulations to overcome policy barriers** | × | **Policy barriers at the neighborhood, site and watershed scale:**<br><br>• **Zoning ordinances**<br><br>  - Article 141 [140] states a minimum of 30 m² free of construction for lots whose area oscillates between 80 m² and 120 m² and are not connected to a sanitary sewer system with a treatment plant. However, if the lot is connected to sanitary sewer system the minimum is 15m² Article 142 [140]. To increase impermeability both areas should count with the same area free of construction<br>  - Article 113 section b [140] states that 50% of the total length of the front of the house most be used as a garage and the other 50% for green areas. However, it does not state regulations on open garage spaces, such as its permeability<br>  - There is no regulation on the roof use for rain-water collection<br>  - There is no regulation on vacant lots for recreation or community gardens, parks, etc.<br>  - There is no regulation on the minimum open space protection<br>  - There is no regulation on the amount of vacant lot needed in case of natural disaster<br><br>• **Roads and transportation**<br><br>  - Article 54 [141] states the configuration of parking spaces within the property. However, none of these configurations refers to permeable parking spaces<br>  - Article 57 [141] states that in industrial and storage areas there may be a complete impermeability coverage. This article should be revised to allow opportunities for permeable options<br>  - There is no regulation on the minimum amount of bicycle paths in the city that could incentivize an urban transportation transition<br><br>• **Construction:**<br><br>  - Article 64 [142] states a minimum width of 50 m within the bank of a surface water body. The NYC GI Plan proposes a 60 m regulations on impervious coverage within the watershed, see Table 2. |

**Table 3.** *Cont.*

| Stage | Policy Insights | Policy Insights Assessment at the Different Scales; See Figure 4. |
|---|---|---|
| **First and second stages Additional policy insights and strategies used by the NYC GI Plan** | × | - Article 586 [142] states that in the absence of road signs indicating the speed limit, will be 50 kph. In NY for urban areas the speed limit is 25 mph or 40.23 kph<br>- There is no regulation on green sidewalks<br>- Even though Chapter 4 states [142] the intention of promoting passive climate control strategies such as green roof, the present regulation states no regulation on the construction of green roofs or policies, such as subsidies for its construction<br>- Article 38 [142] states a mandatory treatment and disposal system that is adjusted to physical conditions of the site to drive wastewater to a sanitary sewer system. However, there is no regulation of how this should be done in a sustainable way<br>- Article 394 [142] should be expanded to include wide sidewalks of 2.4m not only for basic educational use, but also for environmental reasons<br>• **Fractionation, urbanizations and condominiums:**<br><br>- Article 50,129, 175-186 [143] state the assignment and configurations of green areas within the urbanization and condominium. However, there is no regulation on the percentage of permeability of these recreational spaces<br>• **Urban renewal** [143]:<br><br>- There is no regulation that takes into consideration the community involvement for the design of an urban renewal plan<br>• **Municipal code No.7794**<br><br>- Articles 5 and 57 [133] state the existence of citizen participation. However, there is no orientation on how and under what procedure the citizens can participate in the various sectors. This generates a direct impact on the real possibilities of the actors to promote environmental initiatives beyond the denunciation of isolated cases<br>- There is no existence of monetary assessment regulations for the valuation of urban ecosystem services |

**Table 3.** *Cont.*

| Stage | Policy Insights | Policy Insights Assessment at the Different Scales; See Figure 4. | |
|---|---|---|---|
| | **Integration of NBS projects into local municipal goals** | × | **Neighborhood, site and watershed scale:**<br>• Municipal solid waste integral management plan by the Technological Management and Industrial Information Technology Center (CEGESTI) and the environmental management office of the Municipality of Flores [147,148]<br>• Local human development plan of the Municipality of Flores [149]<br>• Conservation and road development of the Municipality of Flores [150]<br>• Plan for the prevention of risks and attention of emergencies of the Municipality of Flores [151]<br>• Regulatory plans of the Municipality of Flores [152] |
| | **Implementation of small-scale demonstration projects to highlight policy barriers and socio-economic incentives** | ✓ | **Neighborhood, site and watershed scale:**<br>• SEE-URBAN-WATER [57] research group will implement demonstration projects in 2020 in collaboration with the Federal Ministry of Education and Research in Germany and the Municipality of Flores<br>• Existing demonstration projects implemented by the Municipality of Curridabat under the "Sweet City" framework (a city modelling approach based on pollination) [153]. Some of these demonstrations include the following: rain gardens, and green sidewalks, among others |

**Table 3.** *Cont.*

| Stage | Policy Insights | Policy Insights Assessment at the Different Scales; See Figure 4. | |
|---|---|---|---|
| **2**<br>**(Re) formulation and adoption** | Advice to regulatory bodies<br>*Reduce policy barriers and increase economic incentives*<br>This paper suggests to the Municipality of Flores the following policies and incentives for the implementation of NBS | ✓ | **Neighborhood scale:**<br><br>● **Incentive mechanism policies:**<br><br>   - Rebates and installation financing policies for homeowners to use environmental practices such as cisterns for water conservation<br>   - Awards and recognition programs related to environmental education, conservation and protection of the water resources<br>   - Grant programs to homeowners interested in sustainable stormwater project and practices<br>● **Retrofit policies:**<br><br>   - Compliance institutional assistance for inter-municipal cross department collaboration between the full spectrum of stakeholders to invest on urban sustainable projects<br>   - Compliance public-private partnership assistance to invest on sustainable solutions<br>   - Regulations on sidewalk's width<br>   - Payment for ecosystem services to support the provision of public services on urban areas<br>   - Regulations on rainwater harvesting<br>● **Emergency and safety codes:**<br><br>   - Implementation hurdles requirements, to face implications on neighborhood liability, pedestrian safety and traffic conditions |

**Table 3.** *Cont.*

| Stage | Policy Insights | Policy Insights Assessment at the Different Scales; See Figure 4. |
|---|---|---|
| | | ✓ | **Site scale:**<br><br>● **Incentive mechanism policies:**<br><br>　　- Price-based instruments as taxes for pollution<br>● **Retrofit policies:**<br><br>　　- Regulations related to the amount of green spaces with recreational amenities in every neighborhood<br>　　- Participatory planning and governance in an effort to get residents involved in the solution<br>　　- Consultation and informative brochures to allow property owners to learn about urban sustainable projects<br>　　- Regulations on biodiversity protection in the neighborhood<br>　　- Green lot retrofit policy for the conversion of vacant lots into green lots<br>　　- Permeable pavement retrofit policy to reduce impervious areas<br>　　- Regulations on cistern and pipping systems<br>　　- Bioretention retrofit policy to treat runoff from private and public property<br>● **Emergency and safety codes:**<br><br>　　- Minimum green area and location in the residential in the events of natural disasters |

**Table 3.** *Cont.*

| Stage | Policy Insights | Policy Insights Assessment at the Different Scales; See Figure 4. | |
|---|---|---|---|
| | Advice to regulatory bodies | ✓ | **Watershed scale:**<br><br>• **Incentive mechanism policies:**<br><br>  - Tax abatement to encourage sustainable economic development urbanization projects<br>  - Subsidies offered to urban planners to use more sustainable designs and green infrastructure practices<br>• **Retrofit policies:**<br><br>  - Regulations related to sustainable urban planning, conservation and protection of urban watersheds<br>  - Policies in the form of ecological fiscal transfer mechanisms creating economic incentives<br>• **Economic instruments:**<br><br>  - Fiscal instrument as policies that incentivize nature-affine investment from public authorities<br>• **Emergency and safety codes:**<br><br>  - Minimum green area and location in the residential in the events of natural disasters |

## 4. Discussion

The process of assessing sustainability policies for the implementation of NBS within the municipal context can be complemented through an urban experiment and an iterative policy feedback cycle (PFC). This experimental framework, known as well as Reallabore, relies upon a series of policy assessment stages that study the current policy context to find barriers and incentives for the implementation of subsequent policy outcomes. In this regard, urban experimentation can contribute to an adaptable policy readiness tool when there is not an existing comprehensive policy guideline for the implementation of new technologies. Hence, this paper proposes a novel method to assess the municipal policy readiness in the implementation of NBS based on the policy insights learned from an existing urban experiment.

Additionally, this novel method aims to contribute to the ongoing debate with respect to the ability of experimentation to prompt scalability and transferability of results. For this purpose, this paper transfers the policy insights of the NYC GI Plan, considered as an urban experiment, to a specific case study. Its objective is to understand if lessons learned from an urban experiment might be transferred to other contexts with similar governance and urban challenges. Here, this paper assesses the policy framework of the NYC and of the Municipality of Flores in Costa Rica as they both suffer from water management and rapid urbanization problems.

Two common transferable lessons or insights seem to arise from the NYC GI Plan as an urban experiment on GI: removing barriers and creating economic incentives. As demonstrated by the PFC of the NYC GI Plan discussed in this paper, to implement urban GI, the first two steps involve determining the most significant policy barriers and incentive programs to target and overcome that barrier. Additionally, the entire policy assessment process yields seven additional policy insights for the successful implementation of policies related to NBS, such as stakeholder participation and integration of NBS project into local municipal goals; see Figure 7.

By applying these policy insights to the case study, in the Llorente district in the Municipality of Flores, a series of policy suggestions were identified. To date, the Municipality of Flores in Costa Rica has implemented a series of regulatory plans to tackle stormwater and rapid urbanization problems. However, these regulations, as discussed in this paper, should capitalize more fully on the multiple benefits provided by NBS. Therefore, it is suggested that the Municipality of Flores revises and reformulates its current regulatory plans to include the full spectrum of stakeholders for decision-making, and to identify possible synergies with other municipal departments to meet local goals using green infrastructures; see Table 3. Here, it is important that the municipality includes in its local policies incentive mechanisms, such as grant programs, rebates and tax abatements for the development of GI, also regulations that enhance the constant knowledge transfer between local citizens and urban planners. Municipalities in this regard can make use of its local funds from stormwater fees and loan programs combined to other institutions funding to stimulate the adoption of GI.

In this case, the proposed PFC in this paper can provide an opportunity to evaluate urban experiments projects related to NBS. Past experiences, in a 27-city study in the USA, demonstrate that the transfer of policy insights using a PFC can reveal both the range of benefits, incentives and difficulties to endorse the implementation of GI [9]. In this regard, municipalities tend to be interested in adopting GI, yet community participation and maintenance costs are still barriers for the further upscaling of GI projects. Therefore, municipalities should focus on the first two stages of the PFC, identification and reformulation of policies to reduce barriers and create socioeconomic incentives.

Additionally, the Municipality of Flores can look at positive examples not only out of its territory, like the case of the NYC, but also in Costa Rica. Currently, the Curridabat Municipality is following a more ecological governance approach. This municipality grants an exception of urban green spaces to enhance pollination, and it encourages its citizens to adopt different types of GI [153]. This has contributed to a more inclusive regulatory plan where citizens can raise their voice and share their knowledge and perceptions to the different officers of the municipality. For example, if a citizen finds a garden that is dirty, he or she can notify this to an official of the municipality via smartphone app. Consequently, other urban experiments that are in the beginning phase, such as the case of the Curridabat Municipality, could also help to promote an understanding of the motivation for NBS and incentivize the application of more inclusive and sustainability policies.

Lastly but not least, it is worth mentioning that assessing policies through the PFC, as a tool to extract policy insights, can be in some scenarios more normative and less practical in decision-making. This becomes evident when the real context is more complex and challenging as expected. For example, in the case study in Llorente, the need to reduce access to the river or to cut trees in front of houses, is viewed as a measurement of local perception of safety rather of recreation. Therefore, assessing the policies through the PFC should also seek to understand more in detailed the local socioeconomic activities, perceptions and stakeholder's interactions to provide a systematic framework needed for decision-making. Therefore, this paper seeks to reduce the weaknesses of the PFC as a policy assessment tool by understanding the reality and externalities of the case study. This is possible by extracting and proposing policy insights relevant to the full spectrum of stakeholders that aim to tackle real social and economic problems.

The reality and externalities of the case study are dependent on its specific context, actors and issues. For instance, it is difficult to apply new policies initiatives when external factors such as controversy among political interests and social perceptions affect the decision-making. In the case of the Municipality of Flores, there is a clear political interest to solve the stormwater and rapid urbanization problem. However, its municipal budget for such goals is very low and the implementation of new projects that impose alternative solutions usually requires long bureaucratic procedures that reduce local interests. Similarly, the governance period, usually four years, becomes a barrier for the development of new policies that require longer socio-economic studies. Therefore, the proposed PFC of this paper takes into consideration the context reality and externalities to highlight possible problems and strategies for urban sustainability policies.

Finally, it may be concluded that the policy assessment of urban experiments can contribute in assessing policies, incentives and barriers that municipalities may have in the implementation of NBS. The solutions to overcome these barriers, such as regulatory and socio-economics one, are often dependent upon the Municipality´s goals and its interaction with other institutions and departments. Institutions with different goals come together to resolve their problems through a common solution to accomplish multiple goals. Additionally, this integrated approach would lead to a more comprehensive policy decision-making system that recognizes the relationship between nature and society, and thus, incentivizing a more sustainable urban transition. In this regard, municipalities such as the one of Flores in Costa Rica should look beyond the compartmentalization of traditional strategies for policy-making and establish opportunities for bottom up approaches. Combined with the constant communication and participation of stakeholders and citizens to provide a more efficient co-design of policies and implementation of alternative solutions. These solutions need to be evaluated and communicated to the different stakeholders (fourth and fifth stage) to star again with the policy cycle where such proposed policies might need to be modified, expanded or even derogated.

## 5. Conclusions

This paper synthesizes urban experimentation and PFC literature to develop an adaptable policy assessment tool on NBS. It´s main goal is to understand urban arrangements, norms, and regulations, and in the process, create a policy readiness tool for the implementation of NBS. This assessment is based on the policy insights learned from the NYC GI Plan as an evidence-based policy making for the replication of related urban experiments. In this regard, the paper argues that an iterative experimental approach can contribute to a sustainable urban transition. That is, urban experimentation can provide the means to highlight both the range of benefits that NBS provide and the policy framework to endorse its implementation. Consequently, this paper extract possible policy barriers and economic incentives from the NYC GI Plan to assess the policy readiness of future projects related to the NBS. One of these projects is focused in managing stormwater and rapid urbanization problems in the Llorente district in the Municipality of Flores, Costa Rica.

The Llorente district in Costa Rica presents socio-economic, environmental and infrastructure practices that pose challenges to an urban planning transition. These challenges can be overcome through experimentation in an urban laboratory that can provide the means for the propagation of sustainable policy incentives and urban designs. However, before implementing NBS in urban areas it is first important to assess the existing policy context and possible barriers for its implementation. Therefore, this study proposes a methodological tool to assess the municipal policy readiness in the implementation of sustainability policies related to NBS. Results suggest that for the implementation of successful NBS at the municipal level, it is important to count with constant stakeholder´s participation, community involvement and technical knowledge transfer. Additionally, municipalities and the full spectrum of stakeholders should take into consideration external trigger events (climatic, socio-economic, environmental, etc.) to review and revise local regulations, and with this, integrate NBS projects into inter-municipal cross departmental goals.

The methodology of this paper relied on the policy assessment tool to extract policy insights of successful urban experiments to assess its policy scalability and transferability to other contexts. This paper recognizes the limitations of this methodology within the social, economic and political contexts. For instance, the policy assessment tool relies on the rational comprehensive criteria where each stage of the policy making is to be evaluated and analyzed [154]. However, in some cases, policy makers can count on limited time and resources, and what is more, with little community involvement and stakeholder participation. Consequently, there are limits in each element of the policy assessment tool. Limits exist in the process, such as lack of knowledge, time, and resources, and limits exist in terms of its outcomes, as new policies create winners and losers and may depend on governing period and political party [154]. However, despite all these limitations, the policy assessment tool signifies an important approach to understand and identify policy barriers and to promote socioeconomic incentives for the implementation of alternative policies and solutions.

**Author Contributions:** Conceptualization, V.A.N. and J.H.; funding acquisition, J.H.; investigation, V.A.N.; methodology, V.A.N. and J.H.; supervision, J.H.; validation, V.A.N. and J.H.; visualization, V.A.N.; writing—original draft, V.A.N., and J.H.; writing—review and editing, J.H. All authors have read and agreed to the published version of the manuscript.

**Funding:** This work was supported by the Federal Ministry of Education and Research in Germany. The funding covered trips to Costa Rica, which was performed by the SEE-URBAN-WATER junior research group (see, https://www.tu-darmstadt.de/see-urban-water/).

**Acknowledgments:** We acknowledge support from the German Research Foundation and the Open Access Publishing Fund of Technische Universität Darmstadt.

**Conflicts of Interest:** The authors declare no conflicts of interest.

# Appendix A

**Table A1.** Laws (in the environmental code), regulations and political constitution of Costa Rica related to the water management.

| | |
|---|---|
| **Organic Law of the Environment (No.7554)** | <ul><li>Article 2: Everyone has the right to enjoy a healthy and environmentally friendly environment, as well as the duty to conserve it [126].</li><li>Article 2: The State will ensure the rational use of the environmental measures to protect and improve the quality of life of the inhabitants [127]</li><li>Article 2: The State is obliged to promote economic and environmentally sustainable development [127]</li><li>Article 6: The State and the municipalities will encourage the active and organized participation of inhabitants in the taking of decisions to protect and improve the environment [127]</li><li>Article 12: The State, municipalities and other institutions, public and private, will promote an environmental culture education to achieve a sustainable development [127]</li><li>Article 17: Every human activity or project that alter or destroy elements of the environment or generate waste, toxic or hazardous materials, will require an environmental impact assessment by the National Environmental Technical Secretariat created in this law (SETENA) [127]</li><li>Article 50,51 For the conservation and sustainable use of water, the following should be applied, among others [127]</li></ul><br>　　a.　Protect, conserve and, where possible, recover aquatic ecosystems and the elements that intervene in the hydrological cycle.<br>　　b.　Protect the ecosystems that allow regulating the water regime.<br>　　c.　Maintain the equilibrium of the water system, protecting each of the components of watersheds. |
| **Law of Biodiversity (No. 7788)** | <ul><li>Article 37: Payment of environmental services (PES). Under programs or projects of sustainability duly approved by the National Council of Areas of Conservation and by the Public Services Regulatory Authority, the Services Regulatory Authority Public may authorize to charge users, through the relevant rate, a percentage equivalent to the cost of the service provided and the size of the approved program or project [128]</li><li>Article 22: The National System of Conservation Areas will be an institutional management and coordination system, decentralized and participatory, in order to dictate policies, plan and execute processes aimed at achieving sustainability in the management of Costa Rica's natural resources [128]</li></ul> |
| **Water Law (No.276)** | <ul><li>Article 149: It is forbidden to destroy, both in national forests and in those of particular, the trees located less than sixty meters from the springs that are born in the hills, or less than fifty meters from those born on land blueprints [129]</li><li>Article 150: It is forbidden to destroy, both in national forests and on land particular, the trees located within five meters of the rivers or streams that run through their grounds [129]</li></ul> |
| **Forest Law (No. 7575)** | <ul><li>Article 33: The following protection areas are declared [130]:</li></ul><br>　　a.　The areas bordering permanent springs, defined in a radius of one hundred meters measured horizontally.<br>　　b.　A strip of fifteen meters in rural area and ten meters in urban area, measures horizontally on both sides, on the banks of rivers, streams or streams, if the terrain is flat, and fifty-meter horizontal, if the terrain is broken. |
| **Law of Health (No. 5395)** | <ul><li>Article 275, 276 and 277: Establish regulations on contamination by solid or liquid waste in the bodies of water and especially in the cases of contamination by trades or industries [131]</li></ul> |
| **The Penal Code** | <ul><li>Article 226: Establishes several norms referring to the irrational exploitation of water, deviation from its channel, excessive use or good, and obstruction for others to use it [132]</li></ul> |

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
