# Peer review of "A Methodology of Policy Assessment at the Municipal Level: Costa Rica´s Readiness for the Implementation of Nature-Based-Solutions for Urban Stormwater Management"

_sustainability, doi:10.3390/su12010230_

Round 1

Reviewer 1 Report

This is very important and timely article based on a good case study from Costa Rica.

In general the article is well justified and based on impressive reference list.

However, there are some issues that need to be addressed.

 I found “Introduction” section is written in a very generic manner. For readers who are not familiar with the specific terminology and concepts of Ecosystem Services, Sustainable development, Green and Grey Infrastructure, Nature Based Solutions and “existing urban experiment” it would be difficult to understand the main goal of this article.  

The main aim of this article is the urban storm management as a particular emphasis on Nature Base Solutions in the case of Costa Rica and the use of New York City Green Infrastructure Plan as a guide for the case study in New York City.  

I would also recommend to include urban stormwater management in the title of the article and to the keywords.
2. Please provide in the “Introduction” all definitions of concepts and terminology (grey and green infrastructure, “traditional infrastructure” (what is it and what are the elements of it? How it is different from the sustainable green infrastructure?). It would be useful to include some concrete examples from literature to support each definition.  Explanation of the Policy Feedback Cycle should also be supported by some examples. Where, in what country and cities it was implemented?

It should be a stronger justification of the appropriateness of using New York City Green Infrastructure Plan as a guiding policy for the Costa Rica case.  Since Costa Rica has a tropical climate, it should be also some differences in the interpretation and implementation of urban water storm management devices.

In the “Background” it should be included some information about climate and ecological peculiarities (e.g. annual rainfall), which are essential for the stormwater management assessment. Location Map of Costa Rica with the indication of the Municipality of Flores would be useful.
Figure 3. Please add some explanation what readers should see in each picture. It would be useful to add another figure: how proposed experimental settings would look like (even on sketch level). It can be similar to Figure 1.

Author Response

Dear Reviewer 1,

We would like to thank you for your constructive comments. Your comments provided valuable insights to refine the content of our paper. Therefore, we tried to address the issues raised as best as possible; see the attached excel file with the complete description and solutions for each comment.

We wouldl like to wish you the very best,

God bless you,

Reviewer 2 Report

- The topic of the paper is an excellent fit for Sustainability.

- The title is informative and very good.

- Abstract is well written, but is, in my opinion, too long. Could you please shorten it to 200 words a required? I recommend to delete some of initial general sentences.

- In the end of the Introduction, could you please shortly present what are individual sections of the paper about? Just to let a reader know, what to expect.

- I carefully read all parts of the paper and have to admitt that the paper is very well written and clearly voices what was meant to say. Theoretical, methodological and empirical parts of the paper are developed professionally. My only reservation is that within the part Discussion, experience from other examples of similar transfers of policies could be mentioned (and referenced). As it stands now, the discussion is without any references, which could support strenght of statements included.

- In the part Conclusion, could you please expand a bit on the limitations of your study?

I think that the paper is very well prepared. My recommendations are rather of technical nature. Let me congratulate to authors for a very nice piece of work.

Kind regards,

Author Response

Dear Reviewer 2,

We would like to thank you for your constructive comments. Your comments provided valuable insights to refine the content of our paper. Therefore, we tried to address the issues raised as best as possible; see the attached excel file with the complete description and solutions for each comment.

We would like to wish you the very best,

God bless you, 

Reviewer 3 Report

This is a well-written article –just a few minor editorial issues;

Page 4 1st line -policy changes within…. Contexts

Page 4 2nd para 2nd line –intervention in order to…

Page 4 2.2.1 2nd para 5th line- is “Major “supposed to be majority?

Page 7 1st para line 7- delete “with this”

Page 8 1st para line 6-“free construction area” is free of building construction?

Page 13 2.3.3 2nd line - is this supposed to be title?

Page 17 next to the last bullet and page 18 2nd para 8th line what is “affine”?

Author Response

Dear Reviewer 3,

We would like to thank you for your constructive comments. Your comments provided valuable insights to refine the content of our paper. Therefore, we tried to address the issues raised as best as possible; see the attached excel file with the complete description and solutions for each comment.

We would like to wish you the very best,

God bless you,
